# Phonon Drag Contribution to Thermopower for a Heated Metal Nanoisland on a Semiconductor Substrate

**DOI:** 10.3390/nano14201684

**Published:** 2024-10-21

**Authors:** Alexander Arkhipov, Karina Trofimovich, Nikolay Arkhipov, Pavel Gabdullin

**Affiliations:** Institute of Electronics and Telecommunications, Peter the Great St. Petersburg Polytechnic University, 195251 St. Petersburg, Russia; karina-khasanova-2001@mail.ru (K.T.); nick10arkh@mail.ru (N.A.); gabdullin_pg@spbstu.ru (P.G.)

**Keywords:** island films, thermoelectricity, phonon drag, confinement effects

## Abstract

The possible contribution of phonon drag effect to the thermoelectrically sustained potential of a heated nanoisland on a semiconductor surface was estimated in a first principal consideration. We regarded electrons and phonons as interacting particles, and the interaction cross-section was derived from the basic theory of semiconductors. The solution of the equation of motion for average electrons under the simultaneous action of phonon drag and electric field gave the distributions of phonon flux, density of charge carriers and electric potential. Dimensional suppression of thermal conductance and electron-phonon interaction were accounted for but found to be less effective than expected. The developed model predicts the formation of a layer with a high density of charge carriers that is practically independent of the concentration of dopant ions. This layer can effectively intercept the phonon flow propagating from the heated nanoisland. The resulting thermoEMF can have sufficient magnitudes to explain the low-voltage electron emission capability of nanoisland films of metals and sp^2^-bonded carbon, previously studied by our group. The phenomenon predicted by the model can be used in thermoelectric converters with untypical parameters or in systems for local cooling.

## 1. Introduction

Field electron emitters based on nanostructured materials are considered promising for the use in vacuum microelectronics devices and for several other applications [1,2]. However, some of them also pose a challenge for physicists, since the physical mechanism that allows electrons to be emitted at room temperature in a relatively weak electric field (of the order of a few V/μm [3,4,5,6]) remains unclear [7]. Nanoisland films of sp^2^ carbon or metal on crystalline Si substrates represent a “model” type of such emitters—the one that is most convenient for experimental study and theoretical consideration. Low-field emission from carbon island films was described in [8,9]. These studies have shown that the films possessed no sharp protrusions, as well as no low-workfunction spots or large workfunction jumps with the inherent “patch fields”, any of which could explain the observed emission capability. Later [10], it was found that films of refractory metals on silicon substrates can also emit electrons in an electric field of approximately the same magnitude. The insensitivity of the effect on the film material indicates that the processes that determine the emission mechanism take place not in the islands themselves, but in adjacent regions of the substrate. In [7,10,11,12], it was suggested that the key process may be associated with a thermoelectric effect, when the thermal energy released in emission centers produces electric fields that facilitate electron emission but disappear thereafter and therefore cannot be detected in subsequent studies of the emitter surface. In [11], a quantitative estimate of the expected magnitude of the thermoelectric effect was given. Yet this estimate was not sufficiently substantiated, since it did not account for the specifics of the thermoelectric effect at the nanoscale. It is well known that a description of kinetic phenomena becomes more complicated when the size of the system approaches the mean free paths of charge carriers and phonons. The phonon mean free path *L*_ph_ in silicon at room temperature can exceed 100 nm, whereas the emitting coatings in [8,9,10] contained nanoislands with a size *d* of the order of 10 nm, and the nanoscale effects should certainly be accounted for.

The field electron emission is usually regarded as “cold”, and nevertheless, nanometer-sized field emission centers can operate in harsh conditions, including relatively high temperature. In our experiments [8,9,10], we often observed optical radiation from emitting spots. In low-current regimes, it had a blue color, typical for luminescence. However, when the extracted current approached the threshold of destruction of an emission center, its color changed to red and then yellow, witnessing the thermal nature of the radiation. This means that in the near-threshold regimes, the centers were heated to 1000 K or an even higher temperature. This observation agrees with literary data on energy distributions of electrons emitted from nanocarbon field-emission structures—the spectra included substantially broadened peaks [13,14]. The corresponding electron temperature was estimated in [15] as ca. 2000 K. Even while the electron temperature in these conditions can be different from the lattice temperature, and the observed broadening can partially originate from other phenomena (e.g., field penetration in the emitting structure), it is clear that the temperature of the centers of “cold” electron emission can amount to at least several hundred Kelvins (albeit, not high enough for notable thermionic emission). For nanosized centers, this also means very high temperature gradients, possibly resulting in high gradients of the thermoelectric potential, which can play a vital role in the emission mechanism.

In this paper, we consider a simple model for maybe a more relevant assessment (in comparison with the one given in our previous work [11]) of the thermoelectric potential generated near an emission center of a nanoisland film or near another heated nanometer-size spot on the surface of a semiconductor.

## 2. Theory

### 2.1. Background

In bulk semiconductors at temperatures of 300 K and above, the main contribution to the thermopower originates from the temperature dependence of the mobility and concentration of charge carriers. Another part is associated with the phonon drag—a phenomenon of transfer of mechanical momentum from atomic vibrations (phonons) to mobile charge carriers. At the macroscale, the drag component dominates only at cryogenic temperatures, when the phonon mean free path is comparable to the size of macroscopic samples [16,17]. At room temperature, and if the doping level is not very high, the probability for a phonon to scatter on other phonons, impurity atoms or defects is much greater than it is for it to scatter on electrons, which reduces the phonon drag. For point contacts of micron- and submicron-scale sizes, the existing theory predicts and experiments confirm further reduction of the drag contribution to thermopower [18,19], as the boundary scattering of phonons “at the contact aperture” is added to the other processes competing with electron-phonon interaction for the mechanic momentum carried by lattice vibrations. However, for yet smaller sizes and in situations characterized by very high temperature gradients, the tendency may change [20,21].

It has been shown [22,23] that the thermal and thermoelectric properties of thin films, nanoparticles and molecules can be controlled by adjusting the position of the Fermi level (by doping or electric field) and hence the charge carrier concentration. In this paper, we will consider the possibility of changing the carrier concentration in a semiconductor near the point contact due to the action of the thermoelectric fields themselves. If under certain conditions the concentration of free carriers can increase significantly in a region smaller than the mean free path of phonons with respect to other scattering channels, this can lead to a notable increase in the phonon drag contribution and thermopower as a whole. The size-related damping of phonon drag will also be taken into account.

We expect the considered effect to be fundamentally nonlinear and gradient-dependent. This excludes the use of the best-developed theoretical models (such as BTE [24,25]) assuming near-equilibrium local phonon and electron distributions and takes into account only the linear terms with respect to deviation from equilibrium—these theories are known to fail when relative gradients of variables are large [26,27,28]. The very general Landauer formalism [20,29] employs potential diagrams, whereas phonon drag forces are non-potential (the effect is also called “the phonon wind”): the integral of the drag force over a closed trajectory can be non-zero. Hence, the following consideration will be based on the most basic physical principles, such as energy and momentum conservation. Our approach also differs from the method used in other studies, where expected manifestations of nonlocality were estimated by evaluating higher-order derivatives for the coordinate dependence of temperature [26].

### 2.2. Problem Statement and Geometry

We will consider the heat flow propagating from a heated metal island into a thick semiconductor plate. The size *d* of the thermal contact between the media has the order of several nanometers or tens of nanometers [8,9,10]. In the semiconductor, the heat is transferred by phonons. Their mean free path *L*_ph_ satisfies the condition *L*_ph_ >> *d* (e.g., refs. [30,31] give the value *L*_ph_ = 210 nm for bulk Si at 300 K). Hence, their propagation is geometrically similar to radial expansion from a point. We will consider only a region of the semiconductor adjacent to the perpendicular to the interface drawn through the center of the contact area, inside a solid angle of 1 steradian. We will assume that conditions are met that allow us to regard the problem in this region as one-dimensional: the motion of charge carriers and phonons occurs in the direction from a single center, and all variables depend on a single coordinate *r*. The contact corresponds to the coordinate *r* = *d* (see Figure 1); it has the temperature *T*_1_ and the electric potential *U*. For simplicity, the contact/semiconductor boundary is ascribed the shape of a spherical segment, with an area equal to *d*^2^. The remote part of the semiconductor plate corresponds to *r*→∞, temperature *T*_2_ < *T*_1_ and zero electric potential.

The metal is ideal, with infinite thermal and electrical conductivity. The heat in the modelled region of the semiconductor is transferred by lattice vibrations only. At temperatures from 300 K and higher, more than 90% of the heat flux is transferred by phonons with wavelengths of the order of a few nanometers [25]. Thus, in a problem with the characteristic scale of the order of 10 nm, the phonons can be regarded as localized particles with certain positions (*r*) in space. In the case of *d* << *L*_ph_, it is necessary to take into account the ballistic (at least partially) nature of heat transfer in the modeled region, which implies that we cannot use the concept of temperature, even a local one [32]. The only exceptions are boundaries: the contact itself (with temperature *T*_1_) and the remote part of semiconductor bulk (*T*_2_). We will consider the phonon population as having two components. One component has an equilibrium distribution corresponding to the bulk temperature *T*_2_. Its parameters are not explicitly included in the equations, but they affect other parameters of the problem: mobility of charge carriers μ, the density of the heat flux through the contact *q*_0_, and the mean free path for the second phonon population component *L*_ph_. This second component is a ballistic unidirectional flow of phonons propagating from the contact into the semiconductor. At the contact, this component carries heat energy flux (or power) *P*_0_ ≡ *P*(*d*) = *q*_0_*d*^2^, and the power decreases with the coordinate as *P*(*r*) = *q*(*r*)*r*^2^ due to scattering processes. Since the region under consideration is small, we will assume that with each scattering event, the interacting “ballistic” phonon “dies” (i.e., is excluded from consideration), and its energy is subtracted from the flux.

For certainty, we take an n-type semiconductor with only one species of mobile carriers—conduction electrons with concentration *n*(*r*) and with immobile donor impurity ions with a constant concentration *n*_D_. Despite the small size of the modeled region, we consider these quantities to be continuously distributed in space. At the equilibrium (for *T*_1_ = *T*_2_), we have *n*(*r*) = *n*_D_ for all positions, and no electric field exists. However, the redistribution of electron density caused by the electron-phonon interaction leads to the appearance of an electric field *E*(*r*), which produces thermoelectric current *I* and/or voltage *U*. In the proposed model, the presence of a velocity distribution of electrons will be ignored—the equations will only include the velocity *v*_e_(*r*) of the “average” electron, with *v*_e_(*r*) = 0 in the equilibrium. When deviating from equilibrium, the local velocity will be determined by electron mobility μ, by the resultant force (the sum of the electrical force −*eE*(*r*) and the force of phonon drag) and by diffusion of electrons against the gradient of *n*(*r*).

### 2.3. Boundary Condition

In [10,11], the heat power generated in the emission center was estimated as 1–2 µW. However, in other situations, it may be necessary to determine the heat flux from the applied temperature difference. Fourier’s law is not applicable at distances *r* < *L*_ph_ where phonon distribution has two components. For such cases, a formula was proposed that is an analogue of the formula for electromagnetic blackbody radiation (see details in [24,31,33]). For distances << *L*_ph_ (to which the temperature difference is applied), it gives the values of heat flux density independent of the distance and much lower than the values predicted by Fourier’s law; even for zero distance, the heat flux density is finite. The “surface brightness” (i.e., the heat flux density per unit solid angle) of a “black” (acoustically matching the medium) metal body with temperature *T* is expressed as follows:(1)$$I_{ph}(T)=\frac{\sigma_{ph}(T)T^{4}}{\pi }.$$

Here, σ*_ph_*(*T*) is the phonon analogue of the Stefan–Boltzmann constant that is temperature-independent only at *T* << *θ*, where *θ* is the Debye temperature for the medium. In general [24,31,33], its value is determined as follows:(2)$$\sigma_{ph}(T)=\frac{3k_{B}^{4}}{8\pi^{2}\hbar^{3}v_{a}^{2}}\int_{0}^{\theta /T}\frac{x^{3}dx}{e^{x}-1}.$$

Here, *k_B_* is the Boltzmann constant, *v_a_* is the speed of sound or, more precisely, the mode-averaged phonon velocity. For the density of the heat flux from a unit area of a metal body at temperature *T*_1_ into a unit solid angle of a semiconducting medium at temperature *T*_2_ (see Figure 1), we can write the following:(3)$$q_{0}=\frac{1}{\pi }(\sigma_{ph}(T_{1})T_{1}^{4}-\sigma_{ph}(T_{2})T_{2}^{4}).$$

Figure 2 shows a plot of this dependence for the Si medium (*v_a_* = 6400 m/s [30,31]), which is very close to linear. Since the calculation was performed for a “black body”, the plotted values are probably overestimated. On the other hand, they are in sensible agreement with the macroscopic formulas. Taking the bulk thermal conductivity of silicon 149 W/m·K and a temperature difference of 100 K applied to a layer of 210 nm (the mean free path of phonons in Si [31], i.e., the minimum distance for which the macroscopic formulas are approximately correct), we obtain a heat flux density of approximately 7 × 10^10^ W/m^2^ or 7 × 10^−8^ W/nm^2^—which matches the value given by the graph in Figure 2 for *T*_1_ − *T*_2_ = 100 K. Hence, we can use formula (3) in our calculations. However, when describing the propagation of the “radiated” phonons, we must account for nanoscale effects by the introduction of additional phonon scattering factors.

### 2.4. Electron-Phonon Interaction

The probability of phonon scattering on electrons is determined by the concentration of the latter *n*(*r*), which can differ from the concentration of the ionized dopant *n*_D_. Let us represent the contribution of electrons to the inverse mean free path of phonons as a product σ*n*(*r*). The coefficient σ has the dimension of area and therefore can be considered as a mean cross-section of the electron–phonon interaction. Classical theory of electron–phonon interaction (see, for instance, in [34]), can be used to estimate its value σ_0_ in the limit *r*→∞, i.e., in equilibrium conditions, in the absence of size effects. In our previous work [11], a convenient intermediate formula can be found that refers to a partial flow of phonons with energy *ћω* and velocity *v_a_*, carrying a heat flux *P_ћω_* through a spot of area *A*. The probability of interaction for a single electron per unit time can be expressed as follows:(4)$$dN_{\hbar \omega -e}/dt=\frac{16\pi \hbar^{3}}{Mm^{2}a{v_{a}}^{3}}\cdot \frac{P_{\hbar \omega }}{\hbar \omega }\cdot \frac{1}{A}.$$

(Compared to [11], some notations have been changed.) Here *M*, *a* and *m* are the atomic mass, lattice parameter and effective mass of the electron, respectively. This relationship can be rewritten as follows:(5)$$dN_{\hbar \omega -e}/dt=\frac{P_{\hbar \omega }}{\hbar \omega }\cdot \frac{\sigma_{0}}{A},$$
where
(6)$$\sigma_{0}=\frac{16\pi \hbar^{3}}{Mm^{2}a{v_{a}}^{3}}.$$

This combination of material characteristics and fundamental constants has the dimension of area and can be interpreted as the cross-section of the electron–phonon interaction. Indeed, the first factor of the right-hand side of the formula (5) is the number of phonons crossing the spot per unit of time. Multiplying it by the ratio of the cross-section σ_0_ to the spot area *A*, we really obtain the frequency of interaction events per electron. It is noteworthy that in the Debye approximation used, the cross-section σ_0_ does not depend on phonon energy and is determined only by material properties.

However, the efficiency of electron–phonon interaction is known to decrease when it is restricted in space [18,19]. In the contemplated model, this property can be described by a coordinate dependence of the parameter σ in the following form:
(7)
σ(*r*) = σ_0_
*s*(*r*),

where *s*(*r*) is a factor accounting for the confinement suppression of electron–phonon interaction. It is natural to assume that *s*(∞) = 1. In the literature [18], we find that the effective degree of electron–phonon interaction for contacts of small size *l* decreases as *l*/*L*_e_, where *L*_e_ is the mean free path of charge carriers. For our region of interest, which has the conical shape with transverse dimension *l* equal to the coordinate *r*, we can describe this factor by the simplest function with suitable asymptotes for *r*→∞ and *r*/*L*_e_→0:
(8)*s*(*r*) = 1 − exp(−*r*/*L*_e_).


However, in the region with strong electric fields (unlike the case considered in [18]), electron motion is also restricted by the distribution of electric potential. Hence, the restricted electron free path can be re-assessed as (*L*_e_^−1^ + *r*^−1^)^−1^. This gives us an alternative expression for *s*(*r*):
(9)*s*(*r*) = 1 − exp[−*r*(*L*_e_^−1^ + *r*^−1^)] = 1 − exp(−1 − *r*/*L*_e_).


(We do not put a factor in front of the exponential to avoid confusing the base of the natural logarithm with the electron charge.)

When choosing between possible representations (8) and (9) of the function *s*(*r*), one can also take into account the fact that in the general case, the question of the relative strength of the electron–phonon interaction in local regions cannot be considered finally resolved—opinions have been expressed in favor of both its dimensional suppression [35,36,37,38] and its enhancement [39,40].

### 2.5. Attenuation of Phonon Flow

We use the “narrow beam approximation” for phonons: all scattered phonons of the ballistic flux are excluded from further consideration, because the scattered phonons most probably acquire sufficient transverse (other than *r*) velocity components to leave the modelled region without the second scattering; thus they carry out their remaining energy and momentum. As the model includes several phonon scattering channels, Matthiessen’s rule requires summation of their contributions to the inverse travel time. However, in the Debye approximation, phonons are characterized by a single velocity *v_a_*, which allows summing the contributions to the inverse path length (−*P*′(*r*)/*P*(*r*))—the prime here and everywhere below means the derivative with respect to the coordinate.

One of the scattering channels is the electron–phonon interaction considered above. Another one is the scattering of phonons of the ballistic flow by phonons of the equilibrium population of the medium. Only U-type (Umklapp) events should be taken into account, since normal (N-) scattering does not change the total momentum carried by the heat flow. The contribution of this channel, together with scattering by defects and impurities, can be accounted for by the value of the phonon mean free path in bulk material *L_ph_*. Assessment of the role of U-type interactions between different phonons of ballistic flow is more complicated. Its contribution obviously depends on the intensity of the flow and therefore is a function of the coordinate. Its upper estimate can be obtained if we take the contact temperature *T*_1_ when specifying *L*_ph_, instead of the bulk temperature *T*_2_. However, this method would most likely significantly overrate this channel’s role. As we know, the phonon drag is predominantly determined by the long-wave part of the acoustic phonon population, and U-processes for these phonons are less probable or completely impossible. Thus, U-interactions within the propagating flow will not be included in the model at this stage, which is best justified for temperatures << θ or for small temperature differences *T*_1_ − *T*_2_.

In nanoscale systems, the phonon mean free path is additionally reduced by boundary scattering. Its contribution for nanowires is estimated by the inverse wire diameter. For the considered conical region (Figure 1), the diameter grows with coordinates and is equal to *r*, so the boundary scattering contribution to the inverse phonon mean free path, by analogy, will be taken as 1/*r*. This can also be interpreted as a consequence of the mismatch between the spherical wave front shape and the concept of phonons as quanta of plane waves of atomic displacements.

The resulting law of attenuation of the ballistic phonon flux propagating into the semiconductor can be given by the following formula:(10)$$P^{\prime }(r)=-P(r)\left(L_{ph}^{-1}+r^{-1}+\sigma (r)n(r)\right)=-P(r)\left(\gamma (r)+\sigma (r)n(r)\right).$$

Here, *γ*(*r*) = (*L_ph_*^−1^ + *r*^−1^) is the attenuation coefficient for all scattering processes except the electron–phonon interaction.

### 2.6. Phonon Drag

Being “heavy” quasiparticles, phonons carry more momentum per unit energy than electrons. According to the theory (for instance, in [41]), the average momentum received by an electron is determined only by the first stage of the phonon scattering process—the absorption of the phonon by the electron (later, the phonon is re-emitted in a random direction; hence, the average value of the momentum it carries is zero). The momentum transmitted per unit of time equals the non-potential force, which eventually produces the thermoelectric potentials and/or current. A formula for the drag force per electron immersed in a unidirectional phonon flow of *P*(*r*)/*r*^2^ flux density has been derived in [11] on the basis of the standard phonon drag theory [34]:(11)$$F_{therm}\approx \frac{16\pi \hbar^{3}}{Mm^{2}a{v_{a}}^{4}}\frac{P(r)}{r^{2}}=\frac{\sigma_{0}}{v_{a}}\frac{P(r)}{r^{2}}.$$

In the Debye approximation (*v_a_* = const(ω)), it does not depend on spectral composition of the phonon flow. In our case, we must account for possible localization effects of substituting *σ*_0_ with *σ*(*r*) (see (6)–(9)). In addition, relation (11) was obtained for a stationary average electron, i.e., for zero macroscopic current. If the average electron has a non-zero velocity *v_el_*, then the additional Doppler factor (1 − *v_el_*/*v_a_*) should be introduced in the formula—so that the average velocity of electrons accelerated by the wave of atomic displacements does not exceed the velocity of this wave:(12)$$F_{therm}\approx \frac{\sigma (r)}{v_{a}}\left(1-\frac{v_{el}}{v_{a}}\right)\frac{P(r)}{r^{2}}.$$

### 2.7. Electric Field and Electric Current

Current *I* must be independent of the coordinate *r* and at position is proportional to the local collective velocity of conduction electrons *v_el_* and to their concentration:(13)$$I=en\left(r\right)v_{el}\left(r\right)r^{2}.$$

Factor *r*^2^ is the cross-section area of the modelled region at position *r*. The current in our model is mainly driven by the phonon drag force, electric field *E*(*r*) and diffusion of charge carriers:(14)$$I=\mu \;F_{therm}(r)\;n(r)\;r^{2}-\mu \;eE(r)\;n(r)r^{2}-e\;D\;n^{\prime }(r)\;r^{2},$$
where *μ* and *D* are the electron mobility and diffusion coefficient, respectively.

We assume that the workfunction values for the metal and the semiconductor are equal, which means the absence of intrinsic contact potentials, charges and polarization associated with chemical heterogeneity. Potential distribution in the system, as it can be demonstrated *a-posteriory*, is very unbeneficial for the appearance of holes, and the holes would also be pushed away by the phonon drag. Thus, the electric field *E*(*r*) originates only from the space charge of electrons with concentration *n*(*r*), with dopant ions with uniformly distributed density *n_D_* and with “mirror reflection” (induced charge) of these charges in the conductive contact. Here we assume that the one-dimensional radial character of potential distribution in the considered region is somehow ensured—by engineering of the material parameter outside the region and/or by selecting appropriate shapes of surrounding electrodes, as is done, for instance, in Pierce-type electron guns. In this case, the relationship between distributions of electric field and electron concentration can be obtained using the Gauss theorem in spherical coordinates with the obvious boundary condition *E*(∞) = 0:(15)$$E(r)=\frac{e}{\varepsilon \varepsilon_{0}r^{2}}\int_{r}^{\infty }(n(\xi )-n_{D})\xi^{2}d\xi .$$

According to this formula, *E*(*r*) is determined by the total electric charge located farther from the coordinate center, at *ξ* > *r*. It would seem more natural to use the charge “inside” (at *ξ* < *r*), but this also includes the induced charge in the contact (island), which is more difficult to determine, while the condition *E*(∞) = 0 means that the absolute values of charges at *ξ* < *r* and at *ξ* > *r* are equal. Knowing *E*(*r*), we can calculate the thermoelectric voltage between the contact and the bulk:(16)$$U=\int_{d}^{\infty }E(r)dr.$$

Substituting (12), (13) and (15) into (14), we come to the following relation:(17)$$I=\frac{\mu \sigma (r)}{v_{a}}(1-\frac{I}{ev_{a}n(r)r^{2}})n(r)P(r)-\frac{\mu e^{2}}{\varepsilon \varepsilon_{0}}\int_{r}^{\infty }(n(\xi )-n_{D})\xi^{2}d\xi \cdot n(r)-eDr^{2}n^{\prime }(r).$$

This integro-differential equation defines the relationship between the sought functions *P*(*r*) and *n*(*r*). The function *σ*(*r*) that is also present here must be explicitly pre-determined by (8) or (9), or in some other form. Current *I* serves as a free parameter, but it can be related to voltage *U* through the properties of the external circuit, for example, its electrical resistance *R* = *U*/*I*. In this case, the current *I* can also be expressed via *n*(*r*)—by means of relation (16).

Equation (10) also connects the functions *n*(*r*) and *P*(*r*). In combination with (17) and with other relations mentioned in the previous paragraph, it gives us a closed system of equations. Its solution, in principle, provides a complete (albeit significantly simplified) quantitative description of phonon drag, a nanocontact, and its contribution to the thermopower. However, the general solution of this system cannot be obtained in analytical form.

### 2.8. Open-Circuit Regime

Analysis of the equations obtained above is greatly simplified for the case of an open external circuit (*I* = 0), when *U* is equal to the thermoEMF. Further simplification is achieved by neglecting the diffusion term in (17)—the post-analysis has shown that its effect on the final result is insubstantial. Then, after reductions, we come to the following:(18)$$\sigma (r)P(r)=\frac{v_{a}e^{2}}{\varepsilon \varepsilon_{0}}\int_{r}^{\infty }(n(\xi )-n_{D})\xi^{2}d\xi .$$

To get this, we divided both sides by *n*(*r*), assuming that *n*(*r*) ≠ 0. However, it should be noted here that *n*(*r*) = 0 also represents a solution, even if a trivial one.

For further numerical calculations, it is convenient to introduce dimensionless variables. Dimensionless power is defined as *W*(*r*) = *P*(*r*)/*β*_0_, where *β*_0_ is a combination of material parameters and fundamental constants:(19)$$\beta_{0}=\frac{v_{a}e^{2}}{\sigma_{0}\varepsilon \varepsilon_{0}}.$$

For this variable, (18) transforms to the following:(20)$$s(r)W(r)=\int_{r}^{\infty }(n(\xi )-n_{D})\xi^{2}d\xi .$$

After derivation, it is as follows:(21)$$s(r)W^{\prime }(r)+s^{\prime }(r)W(r)=-(n(r)-n_{D})r^{2},$$
or
(22)$$W^{\prime }(r)=-\frac{(n(r)-n_{D})r^{2}}{s(r)}-\frac{s^{\prime }(r)}{s(r)}W(r).$$

On the other hand, formula (10) for the dimensionless power is as follows:(23)$$W^{\prime }(r)=-W(r)\cdot \left(\gamma (r)+\sigma (r)n(r)\right).$$

By equating the right-hand sides of (22) and (23) and expressing *n*(*r*), we obtain an algebraic relation between *n*(*r*) and *W*(*r*):(24)$$n(r)=\frac{n_{D}+\frac{W(r)}{r^{2}}\left(\gamma (r)s(r)-s^{\prime }(r)\right)}{1-\frac{\sigma_{0}W(r)}{r^{2}}{\left(s(r)\right)}^{2}},$$
where *γ*(*r*) and *s*(*r*) are explicitly defined known functions. Substituting this into (23), we arrive at the differential equation for *W*(*r*):(25)$$W^{\prime }(r)=-W(r)\frac{\gamma (r)+\sigma_{0}s(r)\left(n_{D}-\frac{W(r)}{r^{2}}s^{\prime }(r)\right)}{1-\frac{\sigma_{0}W(r)}{r^{2}}{\left(s(r)\right)}^{2}}$$

At the first glance, the function *W*(*r*) can be determined by solution of this equation with a boundary condition for the heat flux through the contact *W*(*d*) ≡ *W*_0_ = *q*_0_
*d*^2^/*β*_0_. However, this is correct only for low *q*_0_, below the critical value
(26)$$q_{c}=\frac{\beta_{0}}{\sigma_{0}{\left(s(d)\right)}^{2}}.$$

Otherwise (i.e., for *q*_0_ > *q*_c_), equations (24) and (25) cannot have physically reasonable solutions for the coordinate range *d* < *r* < *r_c_*, where *r_c_* is the positive root of the equation:(27)$$W(r_{c})=\frac{r_{c}^{2}}{\sigma_{0}{\left(s(r_{c})\right)}^{2}}.$$

It is easy to show that *r_c_* > *d* when *q*_0_ > *q_c_*; equity *q*_0_ = *q_c_* corresponds to *r_c_* = *d* when the “critical” region *d* < *r* < *r_c_* narrows to the contact interface.

For *r* > *r_c_*, the denominator in (25) is positive, so the derivative *W*′(*r*) is negative. (For the range of interest, the numerators in (24) and (25) are positive. Otherwise, the situation would be even more complex and interesting than that described below.) Consequently, the function *W*(*r*) representing the residual power of the ballistic phonon flux monotonically decreases with the coordinate. This natural behavior is violated at *r* < *r_c_*. The denominator in (25) is negative here, *W*′(*r*) is positive and the heat power function *W*(*r*) increases with distance from the contact. Meanwhile, the electron concentration takes negative values (as noted earlier, this does not mean the presence of holes)—*n*(*r*) function is described by (24), where the denominator of the right-hand side is the same as in (25). This result should be considered physically invalid. It only means that at *r* < *r_c_*, the balance of forces for electrons cannot be achieved—the drag force here always exceeds the electrostatic force. At the same time, the open-circuit condition *I* = 0 can still be satisfied—by the solution *n*(*r*) = 0, discarded when the formula (18) was derived. It means the absence of mobile charge carriers between *d* and *r_c_*. Thus, the full solution of the considered mathematical problem for electron concentration in the case of *q*_0_ > *q_c_* can be written as follows:(28)$$\left\{\begin{array}{l} n(r)=0,\;for\;d<r\leq r_{c}\\ n(r)=\frac{n_{D}+\frac{W(r)}{r^{2}}\left(\gamma (r)s(r)-s^{\prime }(r)\right)}{1-\frac{\sigma_{0}W(r)}{r^{2}}{\left(s(r)\right)}^{2}},\;for\;r>r_{c} \end{array}\right..$$

Accordingly, the solution for the residual heat flux function *W*(*r*) given by (25) is correct everywhere for *q*_0_ ≤ *q_c_*, but for *q*_0_ > *q_c_*, it is valid only at *r* > *r_c_*. In the carrier-depleted region *d* < *r* < *r_c_*, the function *W*(*r*) is described by a simpler equation, which follows from (23):(29)$$W^{\prime }(r)=-W(r)\gamma (r)\equiv -W(r)\cdot \left(L_{ph}^{-1}+r^{-1}\right).$$

With the boundary condition *W*(*d*) ≡ *W*_0_ = *q*_0_
*d*^2^/*β*_0_, its solution is as follows:(30)$$W(r)=\frac{q_{0}d^{3}}{\beta_{o}r}\exp\left(-\frac{r-d}{L_{ph}}\right).$$

Representing the residual heat flux, function *W*(*r*) must be continuous. Therefore, relations (27) and (30) are valid simultaneously at *r* = *r_c_*:(31)$$W(r_{c})\equiv W_{c}=\frac{q_{0}d^{3}}{\beta_{o}r_{c}}\exp\left(-\frac{r_{c}-d}{L_{ph}}\right)=\frac{r_{c}^{2}}{\sigma_{0}{\left(s(r_{c})\right)}^{2}},$$
or
(32)$$r_{c}^{3}=d^{3}\frac{q_{0}\sigma_{0}{\left(s(r_{c})\right)}^{2}}{\beta_{o}}\exp\left(-\frac{r_{c}-d}{L_{ph}}\right).$$

This algebraic equation allows us to calculate *r_c_*. This parameter (the outer radius of the depleted region) and the value *W*(*r_c_*) ≡ *W_c_* calculated from (31) can further be used to determine the boundary condition necessary to solve Equation (25) at the interval *r* ≥ *r_c_* in the case of *q*_0_ > *q_c_*.

The procedure described above allows us to obtain a numerical solution for the residual heat flux function *W*(*r*). Within the framework of the model under consideration, this is sufficient to assess the phonon drag contribution to the thermoelectric voltage.

### 2.9. Thermoelectric Potential for Open Circuit Regime

The formulas for the potential *U* depend on whether the density of the incident heat flux *q*_0_ is less or greater than the critical value *q_c_*. In the case of *q*_0_ ≤ *q_c_*, the condition (20) (expressing the balance between the electric and phonon drag forces) is valid over the entire coordinate range. Comparing (20) with (15), we arrive at the following:(33)$$E(r)=e\;s(r)W(r)/(\varepsilon \varepsilon_{0}r^{2}).$$

Therefore (see also (16)), the thermoelectric potential can be obtained by integrating the function *W*(*r*), which is the solution of Equation (25), with the boundary condition *W*(*d*) = *P*_0_/*β*_0_:(34)$$U=\frac{e}{\varepsilon \varepsilon_{0}}\int_{d}^{\infty }(s(r)W(r)/r^{2})dr.$$

In the case of *q*_0_ > *q_c_*, the formulas for *E*(*r*) and *U* are different. In the depleted layer (at *r* < *r_c_*), the electron concentration is zero. Hence, for this coordinate range,
(35)$$E(r)=\frac{e}{\varepsilon \varepsilon_{0}r^{2}}\left[\int_{r}^{r_{c}}(-n_{D})\xi^{2}d\xi +\int_{r_{c}}^{\infty }(n(\xi )-n_{D})\xi^{2}d\xi \right]=\frac{e}{\varepsilon \varepsilon_{0}r^{2}}\left[\frac{-n_{D}}{3}\left(r_{c}^{3}-r^{3}\right)+s(r_{c})\;W(r_{c})\right].$$

Taking *W*(*r_c_*) from (31), we can re-write this in another form:(36)$$E(r)=\frac{e}{\varepsilon \varepsilon_{0}r^{2}}\left(\frac{r_{c}^{2}}{\sigma_{0}\;s(r_{c})}-\frac{n_{D}}{3}\left(r_{c}^{3}-r^{3}\right)\right).$$

Farther from the contact (at *r* > *r_c_*), the condition (20) is valid, and the electric field should be determined from (33).

Voltage *U* is calculated by integrating *E*(*r*) over the entire coordinate range. Hence, for the case of high heat flux intensity (*q*_0_ > *q_c_*), we have the following:(37)$$U=\int_{d}^{\infty }E(r)dr=\int_{r_{c}}^{\infty }E(r)dr+\int_{d}^{r_{c}}E(r)dr=\frac{e}{\varepsilon \varepsilon_{0}}\left(\int_{r_{c}}^{\infty }\frac{s(r)\;W(r)}{r^{2}}dr+\int_{d}^{r_{c}}\left[\frac{r_{c}^{2}}{\sigma_{0}\;s(r_{c})\;r^{2}}-\frac{n_{D}}{3}\left(\frac{r_{c}^{3}}{r^{2}}-r\right)\right]dr\right),$$
or:(38)$$U=\frac{e}{\varepsilon \varepsilon_{0}}\left(\int_{r_{c}}^{\infty }\frac{s(r)\;W(r)}{r^{2}}dr+\frac{r_{c}^{2}}{\sigma_{0}\;s(r_{c})}\left(\frac{1}{d}-\frac{1}{r_{c}}\right)+n_{D}\left(\frac{r_{c}^{2}}{2}-\frac{d^{2}}{6}-\frac{r_{c}^{3}}{3d}\right)\right).$$

In this formula, the function *W*(*r*) is the solution of differential Equation (25) with the boundary condition (27).

In the case of *q*_0_ = *q_c_*, the depleted layer is narrowed to zero width at *r_c_* = *d*, and formulas (34) and (38) coincide.

## 3. Results of Numerical Simulations

The behavior of the one-dimensional model for the open-circuit regime described above was tested by numerical simulations, which were performed with the use of Wolfram Mathematica software 14.1. For these early tests, we took material parameters for Si at 300 K [11,31], ignoring their temperature dependence: *L_ph_* = 210 nm, *v_a_* = 6400 m/s, ε = 12, *a* = 0.543 nm, *M* = 28 a.m.u., *m* = 1.08 *m_e_*. This gave as the following the parameters of the model calculated in accordance with formulas (6), (19) and (26): σ_0_ = 8.8 nm^2^, *β*_0_ = 1.75 × 10^−7^ W, *q_c_* = 2.0 × 10^−8^ W/nm^2^. Other model parameters were varied. This included varying the value of dopant concentration *n_D_*, but the results for *U* differed significantly (by more than a few percent) only at extremely high heat flux densities. Below, we show only the data for the lowest doping level *n_D_* = 10^15^ cm^−3^ = 10^−6^ nm^−3^.

Figure 3 shows the results of simulations for two values of contact diameter *d* and three values of the incident flux density *q*_0_. The latter is expressed through its ratio to the “critical” flux density *q_c_*. According to the graph in Figure 1, in the “blackbody” approximation of the value *q_c_* = 2.0 × 10^−8^ W/nm^2^ corresponds to a temperature difference of Δ*T* ≈ 30 K; for real interfaces, this estimate will be greater. Parts (a) and (b) of the figure present the functions of electron density *n*(*r*). Curves 1 and 2 show that at small and moderate values of *q*_0_, the excess electron density is concentrated near the contact. At *q*_0_ > *q_c_* (curve 3), a density maximum with infinite peak value appears, separated from the contact by an electron-depleted region. Over a ca. 10 nm-wide range near the maximum, the electron concentration is greater than 10^−3^ nm^−3^, which means that it exceeds the bulk value *n* = *n_D_* = 10^−6^ nm^−3^ by three orders of magnitude or more.

Parts (c) and (d) in Figure 3 present distributions of the residual power of the ballistic phonon flux *W*(*r*) normalized by its boundary value *W*_0_. The decrease in *W*(*r*) for all the plots in Figure 3c,d occurs at distances significantly shorter than *L_ph_* = 210 nm used in the calculation, i.e., it is mainly determined by scattering on electrons and/or the contribution of the “geometric” factor 1/*r*. Hence, the choice of the *L_ph_* value, within reasonable limits, would not have a substantial effect on the calculation results. Curve 1 corresponds to the minimum value *q*_0_ = 0.1*q_c_*, so the excess electron density (*n*(*r*) − *n_D_*) is relatively low—its contribution to phonon flux attenuation is much less than the contribution of boundary scattering. Notably, at distances below *r_c_*, this curve goes very close to curve 3 representing the data for the highest intensity of phonon flow *q*_0_ = 10*q_c_*. In the latter case, electron density at *r* < *r_c_* is zero, i.e., not far from the low dopant concentration *n_D_* = 10^−6^ nm^−3^. In general, the difference between curve 1 and the other curves in Figure 3c,d reflects the share of the flux absorbed by the excess electrons in each case. For *d* = 10 nm (part (c)), this share is less than for *d* = 30 nm (part (d)), due to the stronger effect of the boundary phonon scattering.

The dependences of the voltage *U* on the heat flux density *q*_0_ (that is, in fact, on the contact temperature difference) are shown in Figure 4a. The calculations were performed for three contact sizes *d*, without taking into account the suppression of electron–phonon interaction due to confinement (we took *s*(*r*) = 1). Yet, the other dimensional effect—boundary (geometrical) scattering of the phonon flux—caused noticeable reduction in the thermoEMF. This conclusion follows from the observed downward shift of the curves with decreasing *d*. As expected, the plots are nonlinear—with complex behavior near *q*_0_
*= q*_c_, when the depletion region appears. The growth of *U* with increasing *q*_0_ is generally not very fast—slower than linear. However, the voltage values predicted by the model are even higher than could be expected. This result will be discussed in the following section.

Figure 5 compares the calculation results for different forms of the function *s*(*r*) describing the confinement effect on electron–phonon interaction—see (8) and (9). The smallest contact size *d* = 10 nm is chosen, since for large sizes, the effect is less pronounced. The upper plot (curve 1) corresponds to *L*_e_ = 0, when the effect is totally disregarded (*s*(*r*) = 1). For the largest value of this parameter *L*_e_ = 100 nm (curve 3), we see reduction of *U* at low *q*_0_ by approximately an order of magnitude. However, when we also took into account the effect of localization on the electrons themselves according to formula (9), the result (curve 4) was close to the result for *L*_e_ = 0.

In Figure 4b, the data of Figure 4a are re-plotted to show the thermoelectric voltage dependence, not on the incident flux density *q*_0_ but on the total power at the contact *P*_0_ = *q*_0_*d*^2^. In this form, the result can be used when considering the problem of the potential of low-field electron emission centers [8,9,10,11], where the boundary condition is set not for the emitting island temperature but the total power released there. The graph in Figure 4b shows that the highest thermoelectric voltage is achieved for the smallest island size. In more detail, the dependence of *U* against the island size is disclosed by the graph in Figure 6. The shown plot corresponds to *P*_0_ = 1 μW—the total power estimates given in articles [10,11]. We expected to find a maximum in the dependency *U*(*d*) corresponding to the highest emission capacity of the islands, but the plot in Figure 6 is monotonous. Analysis shows that with the material parameters and formulas used, the maximum is located at yet lower values of *d*, where the model is apparently irrelevant.

## 4. Discussion

### 4.1. Physical Mechanism

The results of 1d numerical simulations revealed a physical mechanism that can affect the phonon drag contribution to the thermopower in a metal-semiconductor nanocontact. With a sufficiently high temperature difference and a large heat flux through the nanocontact, this phenomenon can prevail over the factors that decrease the drag contribution: namely, the boundary scattering of phonons and the suppression of electron–phonon interaction by confinement. This can be achieved due to a different distribution of electric charge established with the application of a large temperature difference to a nanocontact—in comparison both with a lower temperature difference and with the macroscopic case.

In the macroscopic case, the temperature gradient and the phonon drag force it produces (as well as the action of other mechanisms for creating thermopower) are distributed in the sample approximately uniformly, while the excess charge of free carriers is formed at the sample boundaries or in contact electrodes. Thus, the phonon drag acts on mobile carriers whose concentration is equal to the equilibrium value, i.e., the concentration of dopant ions. If this value is low and the medium temperature is high (non-cryogenic), the scattering of phonons on phonons and impurities is a much more probable process than their scattering on electrons, which makes the phonon drag inefficient.

The performed study showed that the situation changes in the case of nanocontact. The excess concentration of free carriers is formed at a small distance from the nanocontact (see Figure 3a,b), which can be much smaller than the phonon mean free path with respect to scattering on phonons, impurity atoms, and defects. If the temperature difference has the proper sign and sufficient magnitude, the electron concentration in this region can significantly exceed the bulk value. Then the ratio of the probabilities of phonon scattering on phonons and on charge carriers shifts towards the latter, even taking into account additional scattering at the contact boundaries. This can lead to the increase in the phonon drag contribution to thermopower and/or thermoelectric current. Within the framework of the model, at a heat flux density of the order of *q*_c_ (see formula (26)), a concentration of free carriers really achieves the value at which scattering by electrons becomes the dominant scattering process for phonons. Behavior of the *U*(*q*_0_) near *q*_0_ = *q_c_* is complex because of the formation of a layer depleted of the carriers. The influence of this process in the thermoelectric current in regimes with finite external load can be even more dramatic.

Discussing the phonon drag near the nanocontact, we can consider possible analogies with other physical phenomena. One of them is the pressure of acoustic waves, which can be described as a result of the coherent addition of phonons emitted by an oscillating boundary. With increasing distance, the coherence is lost, and the effect transforms into ordinary thermodynamic pressure as a result of the incoherent action of phonons. The electromagnetic analogue is the ponderomotive force acting on charged particles near a metal tip to which an rf potential is applied [42]. This force is determined by the near field, which at short distances has a magnitude significantly greater than the magnitude of the radiated electromagnetic wave. It is quadratic with respect to the field magnitude and pushes charges of any sign out of the region of its concentration—just like phonon drag, which acts on electrons and holes in the same direction. Yet another analogy can be drawn to the thermionic electron emission, with different regimes of current limitation by the space charge and by the temperature.

### 4.2. Relevance of the Model

Numerical modeling performed on the basis of the proposed one-dimensional theoretical model gave us peculiarly high values of thermoelectric voltage. They can be compared with the predictions of classical macroscopic theory as follows. According to previous estimates, the temperature difference Δ*T* = 30 K between the contact and the semiconductor bulk in the blackbody approximation corresponds to a heat flux density through the contact ca. 2 × 10^−8^ W/nm^2^, i.e., equal to *q*_c_ in the model under discussion. For this value of *q*_0_, the model predicts (see Figure 4a) a phonon drag thermoEMF in the range of *U* = 0.5–3.9 V, which corresponds to effective Seebeck coefficients *U*/Δ*T* as high as 17–130 mV/K. This greatly, by one to two orders of magnitude, exceeds the macroscopic values of the Seebeck coefficient even for lightly doped silicon (ca. 0.4 mV/K), as well as the maximum values given by the Mott formula (up to 1.5 mV/K). However, it should be noted that at this stage, the constructed model is based on numerous simplifications, which in most cases lead to an overestimation of the final values. Thus, the presented results of numerical modeling should be considered as upper estimates, which can be useful for assessing the prospects for the implementation of the predicted phenomenon and the need for further research in this direction and improvement of the model.

One of the major simplifications consisted of ignoring the actual distribution of electron velocities and writing the equations for the average electron. In effect, this means adopting a zero value for the Debye radius, a parameter used to account for the thermal velocity spread in electrostatic problems of semiconductor and plasma physics. The *n*(*r*) distributions shown in Figure 3a,b are the narrowest of those obtained in the simulations. They correspond to the bulk concentration of ionized dopant atoms *n*_D_ = 10^15^ cm^−3^ = 10^−6^ nm^−3^. The Debye radius for this concentration and temperature of 300 K is approximately 130 nm, and widening the distribution of *n*(*r*) by this value would obviously change its shape very significantly. However, it should be noted that near the maximum, the carrier concentration here greatly exceeds the value of 10^18^ cm^−3^ = 10^−3^ nm^−3^. If this value is used to calculate the Debye radius, it will be only ~4 nm, and such a broadening of the *n*(*r*) profile will be less substantial. Due to this, neglecting the thermal spread of electron velocities seems justified.

The constructed model also neglects the discreteness of the electric charge, although an electron concentration of 10^18^ cm^−3^ corresponds to only one particle in a 10 nm cube. However, this approach is generally accepted: in quantum mechanical considerations, electrostatic potentials are often associated with a spatially continuous probability density function. A certain error may be introduced by using the blackbody phonon emission formula to describe the properties of the metal/semiconductor interface, but we see no alternatives here.

Some of the sources of errors are obvious and can be taken into account in the next stages of model development. For instance, this concerns the use of material parameters for 300 K in all calculations. The largest value of *q*_0_/*q_c_* = 10 used in the simulations corresponds to Δ*T* = 300 K or more, and material parameters (such as *L_ph_*) should be taken as temperature- or position-dependent. In addition, the largest used contact size *d* = 100 nm hardly satisfies the condition *d* << *L_ph_*, which must also be accounted for. Speaking more generally, the condition for applicability of the 1d model should be formulated more strictly: as *d* << *L*_Σ_, where *L*_Σ_ is the mean free path of phonons with respect to all types of interactions, including the electron–phonon interaction. Obviously, this quantity is position-dependent and can be obtained only by solution of the equations for which it serves as a relevance criterion. However, the situation of the necessity for *a-posteriori* relevance checking is not unusual in physics. An even more serious problem concerns the use of parameters defined for equilibrium conditions in a purely non-equilibrium description.

A significant error in numeric results can be associated with the use of the approximate Formula (6) for definition of the cross-section σ_0_, since the predicted effects are roughly proportional to σ_0_^2^. The formula is based on very bold consideration [11,34] where electron–phonon interaction strength was assessed, simply speaking, from upper estimates of interatomic forces. Hence, the accuracy of this assessment may be rather low. In principle, it could be improved via comparison with the existing data on phonon drag in bulk silicon [17].

The use of a one-dimensional model for description of the process is another serious simplification. In a more rigorous approach, the phonon flux should be considered anisotropic. It is natural to assume that the flux density is maximum near the normal to the contact spot and close to zero near the boundary of the semiconductor plate. Then the largest thermoelectric voltage generated in the direction of the maximum phonon flux density will be shunted by other parts of the contact region with lower flux density. This will lead to the formation of stationary eddy currents [27] with a subsequent decrease in the observed thermoelectric voltage—a similar effect was discussed in [19]. Therefore, a two- or three-dimensional model will most probably show lower values of thermoEMF and current, especially in the presence of surface electrical conductivity along the semiconductor boundary. The thermoEMF predicted by this early one-dimensional model can be realized only with the intentional or accidental fulfillment of special conditions that ensure the elimination of the shunting effect of the boundary.

Summarizing all these apparent objections, we might conclude that the model in its present state can provide only very rough estimates of quantities and maybe some insights into the physics of thermoelectricity on the nanoscale. The above-mentioned sources of error should be given special attention later when developing a more detailed model.

Yet on the other hand, the thermopower coefficient values exceeding the macroscopic ones by several orders of magnitude are in good agreement with the benchmarks given by very different theories for the “ideal” thermoelectric [43,44], for “supernodes” [45], for molecular junctions [46,47,48] and for nanostructures employing quantum interference effects [21,44]. Being much simpler, the suggested model may describe the same “ideal” conditions. In addition, the obtained results agree with the results of works [8,9,10], where high electron emission capability of discontinuous films (presumably enhanced by the thermoelectric effect) was correlated with the presence of islands with a size of ca. 10 nm. The estimates of the thermoelectric voltage obtained in this work are of the order of several volts, which may be sufficient to activate the emission by the “patch field” mechanism. In the case of island films, favorable conditions for producing high thermoelectric potentials could have been realized as a result of a rare combination of random factors, which explains the small number of emission centers observed in [8,9,10].

### 4.3. Prospects of Further Study and Implementation of the Effect

It should be noted that although the proposed model predicts high values of thermoelectric voltage at nanocontacts at large temperature differences, direct experimental observation of the effect may be difficult. The maximum current generated by the mechanism under consideration is small due to the low speed of the produced motion charge carriers, which is limited by the Doppler effect at a level no higher than *v*_a_. Another limitation is related to the depletion layer appearing at high values of heat flux density. Even if the total depletion described by the present model is mitigated in regimes with non-zero current, the charge carrier concentration in this layer is still expected to be low, implying a high internal electrical resistance of the thermoelectric voltage source. Its effect may be particularly strong due to the possible influence of surface conductivity and eddy currents mentioned above. Experimental study of the effect can probably be carried out using an atomic force microscope. Comprehensive description of scanning thermal microscopy method can be found in the classic study [49], and a more recent short review [50] is devoted to local measurements of the thermoelectric coefficient. A good example of the method was given by the work [51], where the absence of size-related suppression of thermopower for point contacts down to 4 nm in size was experimentally demonstrated. Layout and preliminary feasibility analysis for our own planned experiments were disclosed in our previous paper [52].

The next step in our work with the suggested model will consist of numeric simulations of current-voltage characteristics described by Equations (14), (16) and (17) for finite resistance of the external circuit, i.e., for non-zero current regimes. The results of the present work show a drastic increase in mobile carriers’ concentration near the contact for some regimes and depletion for others, which must have an even stronger effect on electric conductivity and current than that on potentials. In further stages of our studies, the basic limitations of the presently over-simplified model will be addressed. We expect that proper regard of two-dimensional or three-dimensional character of the processes (e.g., eddy currents) can substantially reduce the voltage estimates, and the same applies to the more correct definition of the interaction cross-section σ. Both these problems can be solved by two-dimensional simulations using a finite-element software—simulations for a macroscale planar system can help in definition of σ. The equations derived in the present work can be relatively easily adopted to the 2D geometry. However, the statement of the relevant conditions at the semiconductor boundary can represent a serious problem and can strongly influence the results of the simulations.

High internal resistance and low currents of structures based on nanocontacts determine the spheres of their possible implementation. They may be similar to the ones suggested for molecular junctions or supernodes [45,46,47,48], but only if large temperature differences are available. It seems likely that in the near future, they can hardly compete with bulk thermoelectric materials for energy applications. However, there remains the possibility of its use in niche areas where unique (atypical) parameters will be demanded, such as local cooling [45,53].

## 5. Conclusions

In this paper, we considered the possible contribution of phonon drag to thermopower for a special situation of a metal/semiconductor nanocontact with high temperature difference applied. The study was performed to find a plausible explanation for the phenomenon of low-field cold emission of electrons by nanoisland films of metals or carbon on silicon wafers, previously investigated by the authors.

Assuming that in the region adjacent to the heated nanocontact, the phonon distribution is far from equilibrium, we used a very straightforward approach for description of electron–phonon interaction representing both phonons and mobile charge carriers as classical particles. A system of one-dimensional equations for function of phonon flux power and density of conduction-band electrons was suggested and analyzed in detail for the case of an open-circuit regime. Numeric simulations performed on the basis of the developed equations have shown that in these conditions, the electron density near the contact can drastically increase, which results in enhanced contribution of phonon drag to thermopower even at elevated (>300 K) temperatures. The resulting effective thermoelectric coefficients predicted by the suggested model are one to two orders of magnitude greater than its value for macroscopic samples.

In discussion, we agree that the suggested model in its present state may be too over-simplified to provide reliable numeric values of quantities. However, it may be useful for understanding the distinguishing features of thermoelectric phenomena at the nanoscale and can be used as a staring framework for future more advanced studies.

## Figures and Tables

**Figure 1 nanomaterials-14-01684-f001:**
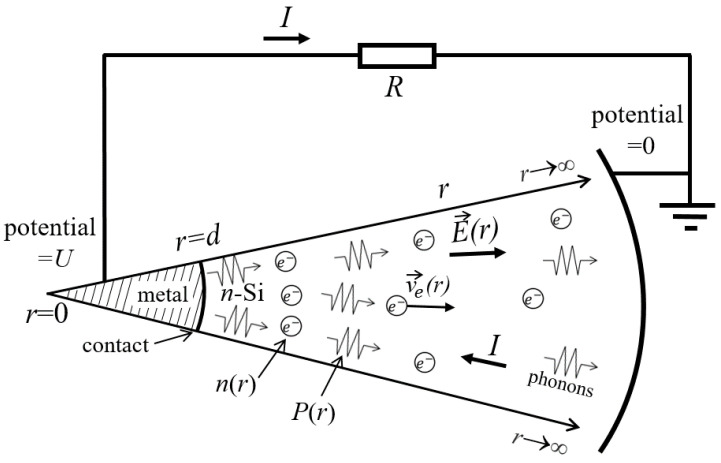
Geometry of the problem. The simulated region in n-Si is the truncated cone within a solid angle of one steradian, with the radial coordinate *r* varying in the range *d* < *r* < ∞. The region *r* < *d* is a metal electrode of the point junction. Arrows show positive directions of electron velocity *v*_e_, electric current *I* and electric field *E*.

**Figure 2 nanomaterials-14-01684-f002:**
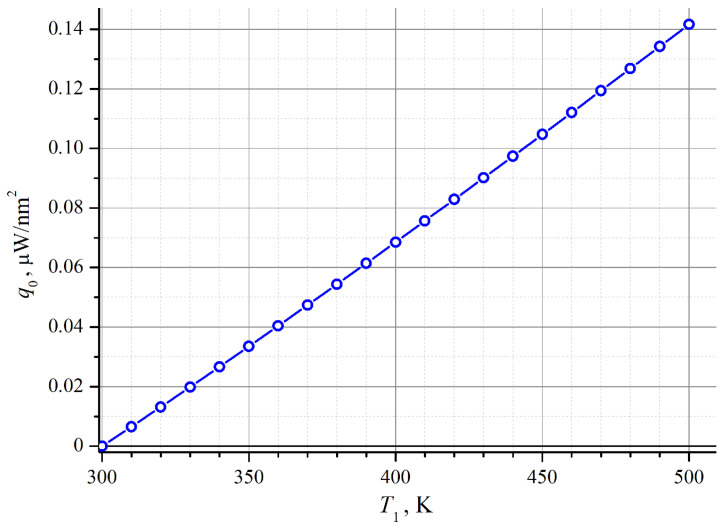
Density of heat flux from a metal body heated to the temperature *T*_1_ into the Si medium at *T*_2_ = 300 K, calculated in accordance with formula (3) for “blackbody phonon radiation”.

**Figure 3 nanomaterials-14-01684-f003:**
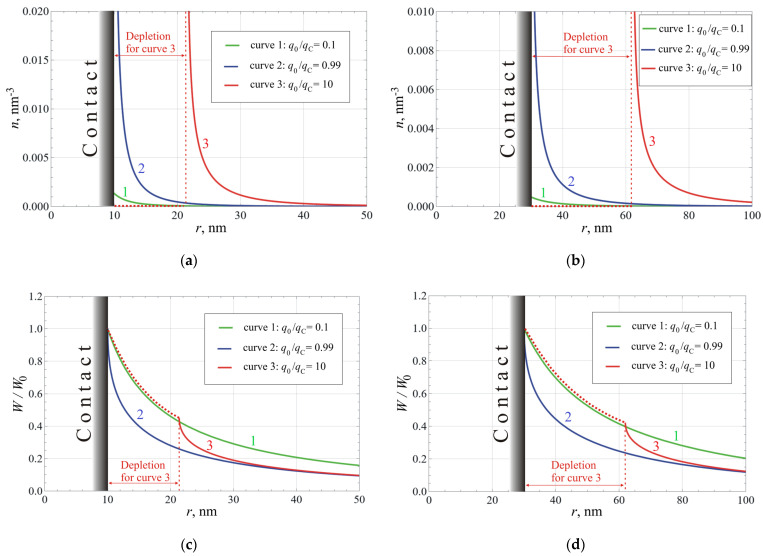
Simulated spatial distributions of the model variables for two values of the contact size *d* = 10 nm (**a**,**c**) and *d* = 30 nm (**b**,**d**), and for three values of incident phonon flux density (see the legends): (**a,b**) Concentration of mobile charge carriers *n*(*r*); (**c,d**) Residual flux of the ballistic phonon flow *W*(*r*) normalized by its value at the contact *W*_0_.

**Figure 4 nanomaterials-14-01684-f004:**
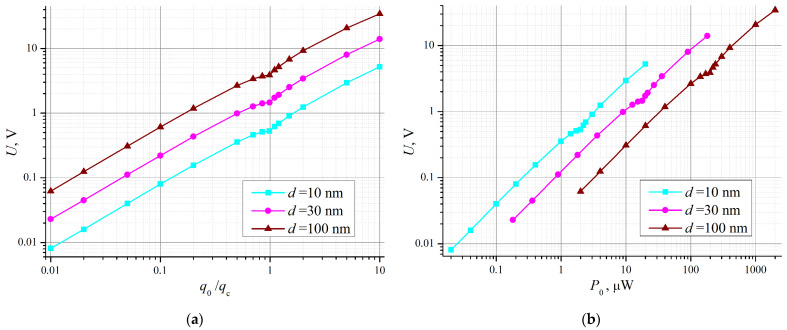
(**a**) Dependency of thermoelectric voltage *U* vs. heat flux density at the contact *q*_0_ calculated in accordance with formulas (34) and (38); (**b**) The same data re-plotted for U as the function full power at the contact *P*_0_ = *q*_0_·*d*^2^.

**Figure 5 nanomaterials-14-01684-f005:**
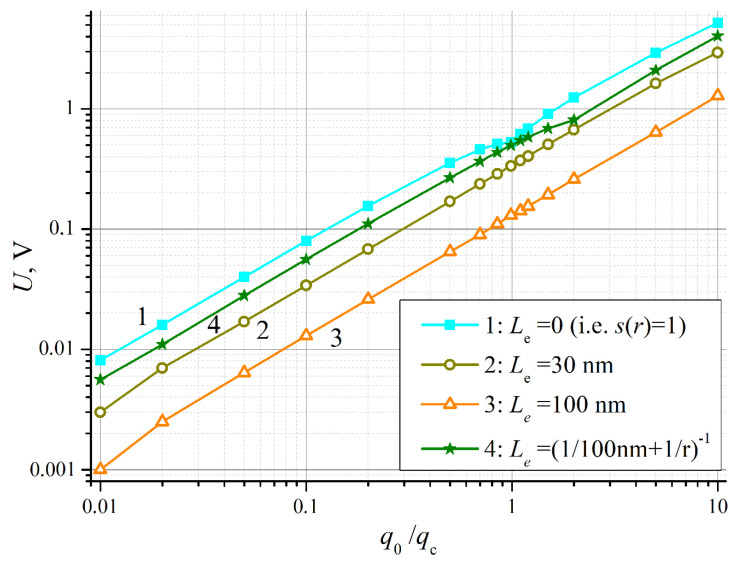
Plots of the thermoelectric voltage *U* vs. heat flux density at the contact *q*_0_ calculated for *d* = 10 nm and different functions *s*(*r*) describing the quenching of electron–phonon interaction due to localization: curve 1—for *s*(*r*) = 1; curves 2 and 3—in accordance with (8) for *L*_e_ = 30 nm and 100 nm, respectively; curve 4—in accordance with (9) for *L*_e_ = 100 nm.

**Figure 6 nanomaterials-14-01684-f006:**
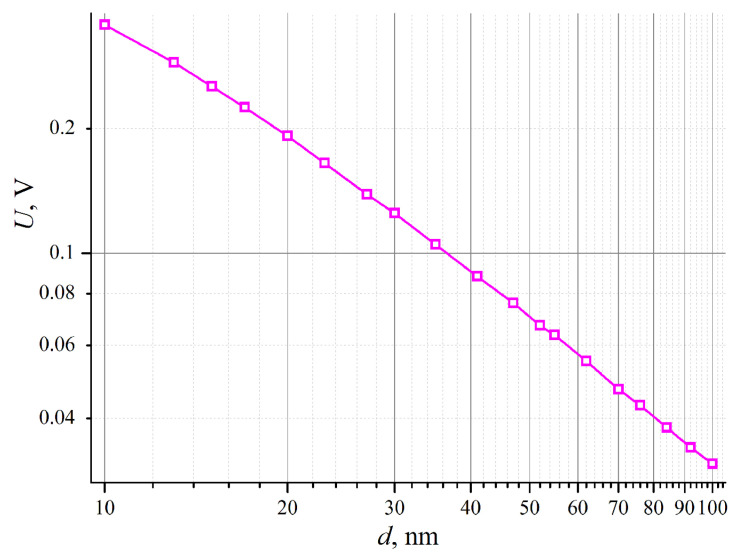
Thermoelectric voltage *U* vs. contact (island) size *d* for a fixed incident heat flux *P*_0_ = 1 μW, calculated with *s*(*r*) = 1.

## Data Availability

The original contributions presented in the study are included in the article; further inquiries can be directed to the corresponding author.

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
