# Peer review of "Phonon Drag Contribution to Thermopower for a Heated Metal Nanoisland on a Semiconductor Substrate"

_nanomaterials, 2024, doi:10.3390/nano14201684_

Round 1
Reviewer 1 Report
Comments and Suggestions for Authors
This paper explores the contribution of phonon drag to the thermoelectric potential of heated metal nanoislands on a semiconductor substrate. The authors consider electrons and phonons as interacting particles based on first principles, and derive the interaction cross-section from the fundamental theory of semiconductors. By solving the equation of motion for average electrons under the simultaneous action of phonon drag and electric fields, they obtain the distribution of phonon flux, charge carrier density, and electric potential. The study finds that despite considering the size suppression of thermal conductivity and electron-phonon interactions, the effects of these factors are smaller than expected. The developed model predicts that a layer with a high charge carrier density will form near the heated nanoisland, which is virtually unaffected by the concentration of dopant ions and can effectively intercept the phonon flow from the heated nanoisland. The calculated thermoelectric potential (thermoEMF) is sufficient to explain the low-voltage electron emission capability of metal and sp2 bonded carbon nanoisland films previously studied. Here are some suggestions:
1. The authors discuss some limitations of the model. It would be beneficial to discuss in detail how these limitations affect the results and propose possible solutions or directions for improvement.
2. The study uses a one-dimensional model. It would be helpful to provide some ideas on how to further develop it into a general model (two-dimensional or three-dimensional model).
3. The authors proposes some potential experimental designs. It is recommended to conduct a corresponding experimental feasibility analysis.
Author Response
Please see the attachment,

Reviewer 2 Report
Comments and Suggestions for Authors
In this paper, the contribution of phonon drag effect to the thermoelectrically sustained potential of a heated nano-island on a semiconductor surface was estimated in a simple one-dimensional model. The developed model predicts existence of a layer with high density of charge carriers which can effectively intercept the phonon flow propagation from the heated island. The effect predicted by the model would be used in thermoelectric conversion local cooling system. The concept of the paper looked attractive and useful for future nano technology, however, the result is still preliminary.
For example, the validity of d<< Lph (p3 line116) where the ballistic nature of heat transfer is taken into account. They cannot determine the value “d” within the flamework of this model, but need another simulation for the estimation. This kind of estimation is important to check the applicability of the model. Another example is the temperature setting is too sharp. The temperature difference of several hundred degrees (Figure2) between nano-island and the semiconductor seems difficult to accept, leading to the unreasonable heat flow. The situation is dynamical. The temperature distribution should be included in this kind of model even though the simplicity would be lost.
Since the model is too simple, I think it is not necessary to propose the actual experimental substance.
I like the idea of one-dimensional model and the phonon drag, therefore, I think the paper will be reformed and provide more useful information to the readers of the Journal.
Round 2
Reviewer 2 Report
Comments and Suggestions for Authors
I think the paper is now ready to publish.